# Three-Year Longitudinal Study: Prevalence of *Salmonella Enterica* in Chicken Meat is Higher in Supermarkets than Wet Markets from Mexico

**DOI:** 10.3390/foods9030264

**Published:** 2020-03-02

**Authors:** Iván D. Regalado-Pineda, Rene Rodarte-Medina, Carolina N. Resendiz-Nava, Cinthia E. Saenz-Garcia, Pilar Castañeda-Serrano, Gerardo M. Nava

**Affiliations:** 1Departamento de Investigación y Posgrado en Alimentos. Universidad Autónoma de Querétaro, Querétaro, QRO 76010, Mexico; david_fc80@hotmail.com (I.D.R.-P.); rodarte.rene@hotmail.com (R.R.-M.); carolina.resendiz.90@gmail.com (C.N.R.-N.); elizabeth.saenz.garcia@gmail.com (C.E.S.-G.); 2Centro de Enseñanza, Investigación y Extensión en Producción Avícola. Universidad Nacional Autónoma de Mexico, Tláhuac, CDMX 13300, Mexico; pilarcs@unam.mx

**Keywords:** *Salmonella*, chicken meat, supermarkets, wet markets, prevalence

## Abstract

Worldwide, chicken meat is considered one of the main sources of *Salmonella enterica* in humans. To protect consumers from this foodborne pathogen, international health authorities recommend the establishment of continuous *Salmonella* surveillance programs in meat. However, these programs are scarce in many world regions; thus, the goal of the present study was to perform a longitudinal surveillance of *S. enterica* in chicken meat in Mexico. A total of 1160 samples were collected and analyzed monthly from 2016 to 2018 in ten chicken meat retailers (supermarkets and wet markets) located in central Mexico. The isolation and identification of *S. enterica* was carried out using conventional and molecular methods. Overall, *S. enterica* was recovered from 18.1% (210/1160) of the chicken meat samples. Remarkably, during the three years of evaluation, *S. enterica* was more prevalent (*p* < 0.0001) in supermarkets (27.2%, 158/580) than in wet markets (9.0%, 52/580). The study was 3.8 times more likely (odds ratio = 3.8, *p* < 0.0001) to recover *S. enterica* from supermarkets than wet markets. Additionally, a higher prevalence (*p* < 0.05) of this pathogen was observed during the spring, summer, autumn, and winter in supermarkets compared with wet markets. Moreover, the recovery rate of *S. enterica* from supermarkets showed a gradual increase from 20.78% to 42% (*p* < 0.0001) from 2016 to 2018. Interestingly, no correlation (*p* > 0.05) was observed between the *S. enterica* recovery rate in chicken meat and reported cases of *Salmonella* infections in humans. Higher levels of *S. enterica* in chicken meat retailed in supermarkets are not unusual; this phenomenon has also been reported in some European and Asian countries. Together, these results uncover an important health threat that needs to be urgently addressed by poultry meat producers and retailers.

## 1. Introduction

Infections caused by *Salmonella* remain an important threat for human health. It has been estimated that, worldwide, this pathogen causes ~20 million human cases and ~140,000 deaths per year [1]. In Mexico, the National Epidemiological Surveillance System reports more than 110,000 *Salmonella* infections in humans every year [2]. Importantly, raw chicken meat is considered to be one of the main sources of *Salmonella* for humans [3,4,5], and it was estimated that ~30% of foodborne Salmonellosis worldwide could be linked to poultry meat [6].

In some countries, it is of particular interest to perform analysis of *S. enterica* in supermarkets and wet markets (places dedicated to sell fresh meat, fish, and produce, aka public markets) due to the significant differences in sanitary conditions between these two retail places [7,8,9,10,11]. Wet markets are an important source of affordable food; unfortunately, these places have been linked to major outbreaks of diseases due to poor hygiene conditions [12,13,14,15]. In fact, a higher prevalence of foodborne pathogens in wet markets compared to supermarkets has been documented in many countries [16,17,18,19]. Importantly, various studies have identified wet markets as an important source of *S. enterica*, where prevalence in chicken meat ranged between 20% and 75% [20,21,22,23]. In fact, some studies have reported higher levels of *Salmonella* contamination in poultry meat sold in wet markets compared to supermarkets [18,20,24,25]; however, a few studies have also reported opposite trends [26,27,28,29]. Unfortunately, these types of analyses are scarce in Mexico; thus, the aim of the present study was to analyze the prevalence of *S. enterica* in chicken meat retailers (wet markets and supermarkets) located in central Mexico.

## 2. Materials and Methods 

### 2.1. Sample Collection and Microbiological Analysis

The analysis was carried out in chicken meat retailers, five wet markets, and five supermarkets, located in Central Mexico (Queretaro State, Lat Long = 20.588793, −100.389885). Every month for three years, from January 2016 to December 2018, a total of 1160 meat samples were collected from wet markets and supermarkets. These meat retailers distribute defeathered and eviscerated chicken carcasses obtained from commercial poultry processing plants. Samples were transported on ice to the laboratory for processing within 4 h. The number of samples per type of market are described with detail in Appendix A. Briefly, in 2016, a total of 680 samples were collected and analyzed. In 2017 and 2018, 240 samples per year were collected and analyzed.

The isolation of *S. enterica* was performed as described elsewhere [12]. Briefly, each sample consisted of 25 g of skin, obtained from one leg and one thigh, homogenized in 225 mL of buffered peptone water (BPW) and incubated at 37 °C for 24 h. For *Salmonella* enrichment, 0.1 and 1.0 mL aliquots of incubated-BPW were transferred to 10 mL of Rappaport-Vassiliadis and Tetrathionate plus iodine solution broth and incubated at 42 and 37 °C for 24 h, respectively. A loopful of the culture was streaked onto xylose lysine deoxycholate agar (XLD) supplemented with sodium novobiocin (0.001% *w*/*v*) and incubated at 37 °C for 24 h [30,31]. At least three presumptive *Salmonella* colonies were streaked on Trypticase Soy agar to obtain pure cultures and then subjected to a urease test. The identification of *Salmonella* isolates was performed using PCR assays.

### 2.2. Identification of S. enterica by PCR Assays 

The DNA samples from presumptive *Salmonella* isolates were subjected to PCR amplification using two *Salmonella*-specific assays targeting *invA* (primer forward: CTGTTGAACAACCCATTTGT and reverse: CGGATCTCATTAATCAACAAT) [32] and *16S rRNA* genes (primer forward: ACGGTAACAGGAAGMAG and reverse: TATTAACCACAACACCT) [33]. PCR *invA* amplification (~437 bp.) consisted of an initial denaturation step of 3 min at 94 °C and 35 cycles of 45 s at 94 °C, 30 s at 57.4 °C, and 30 s at 72 °C, followed by a final extension of 5 min at 72 °C. The protocol for *16S rRNA* amplification (~402 bp.) was similar, except that 32 cycles of 20 s at 94 °C, 30 s at 53 °C, and 30 s at 72 °C were used. PCR products were analyzed on 1.5% agarose gels stained with ethidium bromide.

### 2.3. Temperature, Precipitation, Chicken Meat Production, and Human Cases Correlation Analysis

To uncover relationships between local weather conditions, amounts of chicken produced, and reported human Salmonellosis cases, official databases were consulted and data from 2016 to 2018 were retrieved, archived and analyzed. Monthly local temperatures were obtained from the National Meteorological Service (SMN, for its acronym in Spanish) [34]. Monthly chicken meat production was obtained from the National Service of Health, Food Safety (SENASICA, for its acronym in Spanish) [35], and monthly *Salmonella* human cases from the Weekly Epidemiological Bulletin—Secretary of Health (SSA, for its acronym in Spanish) [2].

### 2.4. Statistical Analysis

The results of *S. enterica* prevalence between years, seasons, and retailers were compared by the Chi-square test [19,36] using XLSTAT software. Confidence intervals (95%) for proportions were calculated using the Wilson procedure with a correction for continuity as described elsewhere [37]. Odds ratios (OR) and 95% confidence intervals were calculated using MedCalc Software. The Pearson correlation coefficient and ANOVA (Tukey post hoc test) using temperature, precipitation, chicken meat production and human Salmonellosis cases data were also performed with the XLSTAT software. Differences were considered significant at *p* < 0.05. Combined odds ratios (Synergy Factor) were estimated as described elsewhere [38].

To corroborate that sample number differences between years do not generate different outcomes, additional analyses were performed using a comparable number of samples (*n* = 240) per year. Briefly, a subset of samples (*n* = 20 per month) was selected from the whole 2016 sample collection using a random number generator (XLSTAT software). Statistical analyses were performed as described above. 

## 3. Results and Discussion

Overall, *S*. *enterica* was recovered from 18.1% of the 1160 raw chicken meat samples analyzed. The prevalence of this pathogen increased (*p* < 0.001) over the three-year period evaluated, from 13.7% in 2016 to 27.1% in 2018 (Table 1). Comparable levels of *S. enterica* contamination (21.3%) in chicken meat retailers have been observed in cross-sectional studies performed in the North, Central, and South regions of Mexico [39,40]. Additionally, this *S. enterica* prevalence in chicken meat has been observed in other world regions such as Australia, Belgium, Brazil, Canada, China, Colombia, Ecuador, Portugal, Spain, USA, Venezuela, and Wales, where contamination levels ranged between 9.5% and 65.0%, [19,26,27,41,42,43,44]. These results indicate that chicken meat retailed in markets could represent an important risk factor for *Salmonella* infections in humans. 

The longitudinal design allowed us to perform a robust statistical assessment [45] of *Salmonella* contamination levels over different seasons. A few studies, from different world regions, have examined the presence of this pathogen in chicken meat retailers over one year period or longer, and some of them have reported discordant results regarding seasonal trends [8,23,27,29,44]; therefore, it is fundamental to perform multi-year analyses to corroborate temporal patterns of *S. enterica* contamination in chicken meat. After three years of microbiological examination, *Salmonella* isolation rates ranged from 8.7% to 30.0% per month and 15.1% to 20.9% per season. Nevertheless, statistical assessment revealed no significant (*p* > 0.05; odds ratio >1.5, 95% CI: 1.0–2.3) differences in the prevalence of *S. enterica* between months or seasons (Table 1). Interestingly, analysis of a single-year showed a higher (*p* < 0.01) isolation rate (20.3%) in winter 2016 compared with the rest of the seasons; however, this tendency was not observed in 2017 (15.0%) and 2018 (28.3%) (Table 2), suggesting that seasonal trends may depend on other factors rather than seasonal conditions solely. To corroborate this idea, local seasonal conditions and poultry meat production were analyzed. No statistical relationships (*p* > 0.05) between temperature, precipitation, chicken meat production, and *Salmonella* prevalence in chicken meat were observed (data not shown). Moreover, another multi-year study revealed that seasonal differences in *S. enterica* prevalence are year-dependent [46]. Overall, these results support the lack of seasonal trends in *S. enterica* meat contamination and could help to explain the discordant seasonal trends reported in the literature [8,28,29,36,44]. Moreover, these results highlight the importance of performing a multi-year analysis for *S. enterica* seasonality assessment.

Remarkably, the present multi-year study also revealed repeatable higher *S. enterica* contamination levels in chicken meat retailed at supermarkets than in wet markets. Overall, the recovery rate of *S. enterica* was higher (*p* < 0.0001) in supermarkets (27.2%, 158 / 580) compared to wet markets (9.0%, 52/580) (Figure 1A). The study showed that it was 3.8 times more likely (odds ratio = 3.8, *p* < 0.0001) to recover *S. enterica* from supermarkets than wet markets (Table 3). In the majority (9/12) of the months analyzed, it was 3.7 to 29.4 times more likely (*p* < 0.019) to recover *S. enterica* from supermarkets than wet markets (Table 3). Additionally, in every year analyzed, from 2016 to 2018, the prevalence of this pathogen was higher (*p* < 0.001) in supermarkets (20.8%, 30.8%, and 41.7%, respectively) than wet markets (6.4%, 12.5%, and 12.5%, respectively) (Figure 1B); it was, at least, 3.1 times more likely (*p* < 0.019) to recover *S. enterica* from supermarkets than wet markets in the analyzed period (Table 3). Moreover, higher (*p* < 0.05) *Salmonella* contamination levels were observed in spring, summer, autumn, and winter in supermarkets (ranging from 23.1% to 36.6%) compared with wet markets (ranging from 7.1% to 13.5%) (Figure 1C); the analysis showed that it was between 1.9 and 10.5 times more likely (*p* < 0.05) to recover *S. enterica* from supermarkets than wet markets during each of the four seasons evaluated (Table 3). Interestingly, analysis of combined factors revealed a higher risk (combined odds ratio = 2.8, 95% CI: 4.5–24.2, *p* = 0.028) of *Salmonella* contamination when chicken meat was acquired in supermarkets during winter season compared to meat from wet markets (Table 4). Finally, to corroborate that results obtained in the present study were not influenced by differences in sample number per year (2016, 2017, and 2018), all statistical assessments were repeated using a similar number of samples (*n* = 240) per year. No significant differences between the two analyses were observed (*Appendix A*). After three years of microbiological examination, it was observed that the prevalence of *S. enterica* in chicken meat retailed at supermarkets increased (*p* < 0.0001) 20.9% from 2016 to 2018. Importantly, our research group detected comparable contamination trends in markets from five other Mexican states, in which *Salmonella* recovery rates were higher (*p* = 0.009) in supermarkets (9.9%, 17/171) and wet markets (3.6%, 9/247) (*unpublished results*). Together, these results indicate that higher levels of *Salmonella* contamination in supermarkets were not a year-dependent phenomenon.

Higher rates of *Salmonella* contamination in chicken meat retailed in supermarkets is a surprising result because superior sanitary and quality standards are expected at these types of stores [7,8]. However, numerous reports from European and Asian countries have showed higher or comparable *Salmonella* contamination levels in meat samples from supermarkets when compared with wet markets samples [8,23,26,27,28,29,44]. For example, studies performed in Spain and Russia showed a higher (*p* > 0.05) *Salmonella* recovery rate (75.0% and 60%, respectively) in supermarkets than wet markets (25.0% and 7.7%, respectively) [26,47]. Additionally, studies from other world regions have reported no difference (*p* > 0.05) in *Salmonella* contamination levels in meat between these two types of retail markets [23,27,28,29]. The high frequency of *Salmonella* in chicken meat from supermarkets could be explained, in part, by the extended shelf life (~6 days) in these types of stores [48], compared with the few hours (>16 h) in which the product is traded in wet markets [26]. Nonetheless, further studies are required to identify the cause of this phenomenon. Taken together, these results uncover an important health threat that needs to be urgently addressed by poultry meat producers and retailers.

To gain insights into the contribution of chicken meat contaminated with *Salmonella* to human Salmonellosis in Mexico, we performed regression analysis using monthly prevalence results (present study), and the monthly number of human Salmonellosis cases reported from 2016 to 2018 by the Weekly Epidemiological Bulletin – SSA. According to this epidemiological report, the monthly prevalence rate of human Salmonellosis was higher (*p* > 0.05) in 2016 (average = 1.6 cases per 100,000 people), followed by 2017 (1.1 cases per 100,000 people), and 2018 (0.6 cases per 100,000 people). Interestingly, no statistical relationships (*p* > 0.05, R^2^ < 0.13) were observed between overall *Salmonella* prevalence in chicken meat and human Salmonellosis cases reported between 2016 and 2018 (Figure 2A). Likewise, no relationships (*p* > 0.05, R^2^ < 0.33) were observed between *Salmonella* prevalence in chicken meat retailed at supermarkets or wet markets and human Salmonellosis cases reported during the evaluated period (Figure 2B,C). Other reports have documented this lack of association between *Salmonella* contamination in chicken meat and human cases [41,46]; however, we believe that there is not enough evidence to rule out chicken meat as an important source of *Salmonella* for humans in Mexico [4]. *Salmonella* foodborne transmission is a complex biological trait and *Salmonella* meat prevalence data may not be enough to establish epidemiological links [4]. Thus, further and comprehensive studies, such as virulence or genomic profiling, are required to elucidate the contribution of chicken meat to the Salmonellosis cases in humans.

## 4. Conclusions

The overall prevalence of *S. enterica* was 18% in chicken meat retailed in markets from central Mexico. Interestingly, *Salmonella* contamination levels were consistently higher in supermarkets than in wet markets. Additionally, it was revealed that the prevalence of this pathogen in chicken meat has increased in the last three years in this Mexican region. Together, these results uncover an important health threat that needs to be urgently addressed by poultry meat producers and retailers. We hope that these data serve as a framework for the poultry industry, policy makers, and health authorities to establish effective *Salmonella* control programs. 

## Figures and Tables

**Figure 1 foods-09-00264-f001:**
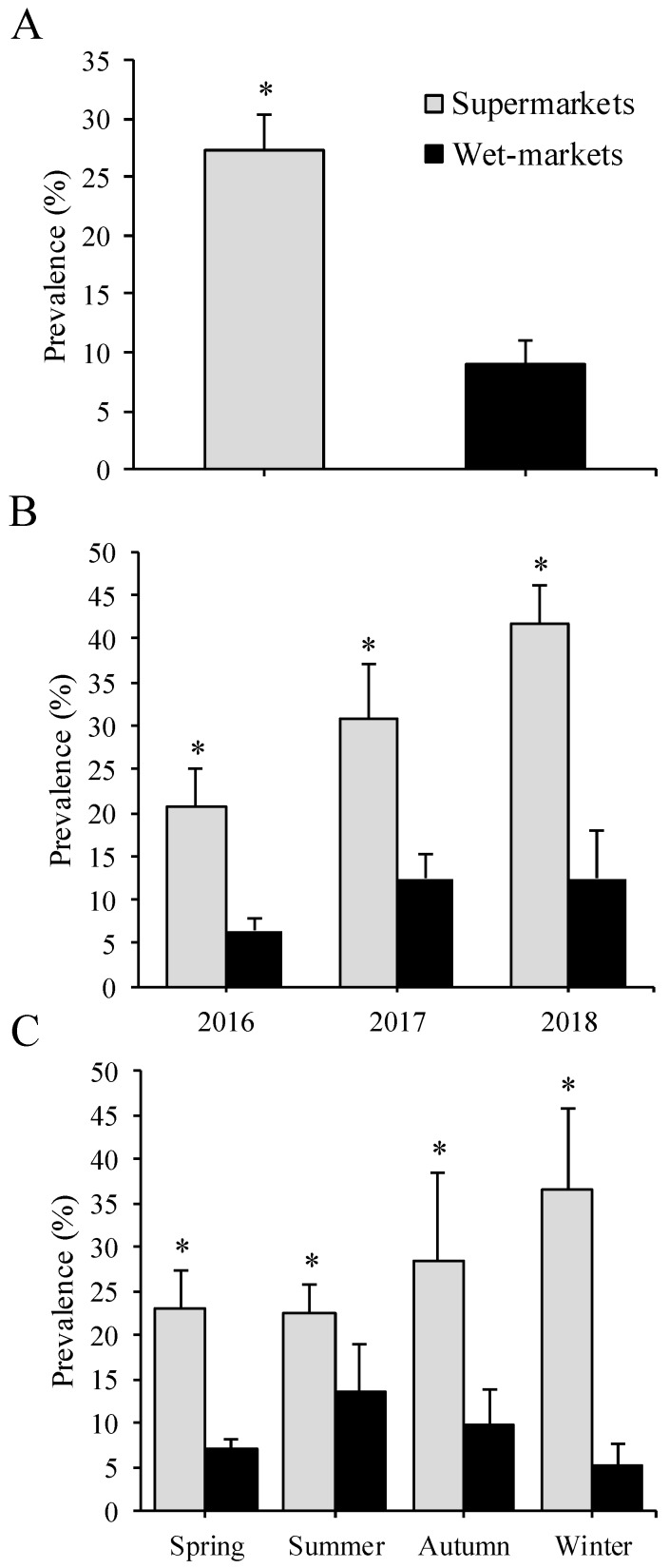
*Salmonella* prevalence in chicken meat influence by market type (**A**), year (**B**) and season (**C**). Asterisks indicate statistical differences (Chi-square test; *p* < 0.05) between supermarkets and wet-markets.

**Figure 2 foods-09-00264-f002:**
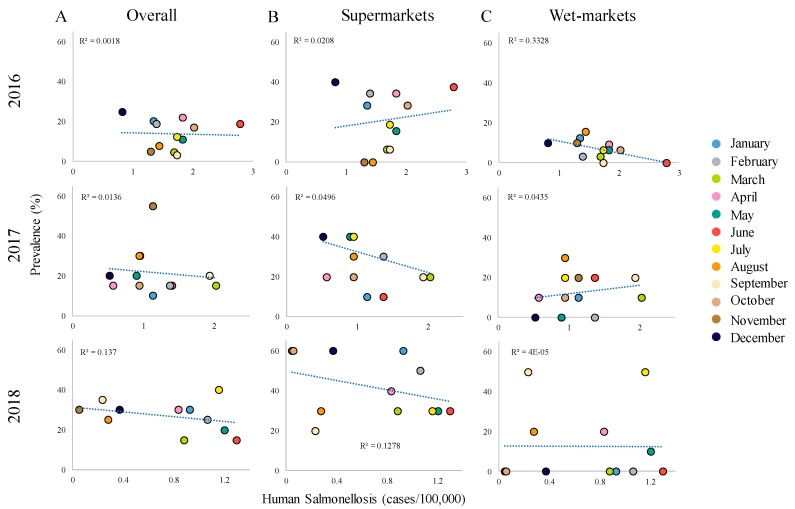
Correlation between *Salmonella* prevalence in chicken meat retailers and human salmonellosis cases. Analyses were performed using overall (**A**), supermarkets (**B**), wet-markets (**C**) *Salmonella* prevalence in chicken meat versus number of Salmonellosis cases.

**Table 1 foods-09-00264-t001:** *Salmonella* prevalence over time.

	No. of Samples	Prevalence (%)	95% CI
**Overall**			
2016–2018	1160	18.1	15.9–20.5
**Annual**			
2016	680	13.7	11.2–16.6
2017	240	21.7	16.7–27.5
2018	240	27.1	21.7–33.3
**Seasonal**			
Spring	312	15.1	11.4–19.6
Summer	312	17.9	14.0–22.8
Autumn	268	19.0	14.6–24.4
Winter	268	20.9	16.3–26.4
**Monthly**			
January	104	20.2	13.2–29.4
February	104	19.2	12.4–28.4
March	104	8.7	4.3–16.2
April	104	22.1	14.8–31.5
May	104	14.4	8.6–23.0
June	104	17.3	10.9–26.3
July	104	21.2	14.0–30.5
August	104	15.4	9.3–24.1
September	104	12.5	7.1–20.8
October	104	19.2	12.4–28.4
November	60	30.0	19.2–43.4
December	60	25.0	15.1–38.1

**Table 2 foods-09-00264-t002:** Seasonal effect on *Salmonella* prevalence per year.

Season	2016	2017	2018
Spring	24/192 (12.5) ^b^	10/60 (16.7) ^a^	13/60 (21.7) ^a^
Summer	25/192 (13.0) ^b^	15/60 (25.0) ^a^	16/60 (26.7) ^a^
Autumn	14/148 (9.4) ^b^	18/60 (30.0) ^a^	19/60 (31.7) ^a^
Winter	30/148 (20.3) ^a^	9/60 (15.0) ^a^	17/60 (28.3) ^a^

Columns with different letter are statistically different (Chi-square test; *p* < 0.05).

**Table 3 foods-09-00264-t003:** Odds ratio analysis of the *Salmonella* prevalence in supermarkets over time.

	Odds Ratio *	95 % Confidence Interval	*p* Value
**Overall**			
2016–2018	3.8	2.7–5.3	*p* < 0.0001
**Months**			
January	4.2	1.4–12.5	*p* = 0.0104
February	29.4	3.8–229.9	*p* = 0.0013
March	3.9	0.8–19.7	*p* = 0.1008
April	3.7	1.3–10.4	*p* = 0.0123
May	4.9	1.3–18.6	*p* = 0.0194
June	11.1	2.4–51.4	*p* = 0.0021
July	1.6	0.6–4.1	*p* = 0.3390
August	0.5	0.2–1.6	*p* = 0.2815
September	0.8	0.3–2.7	*p* = 0.7670
October	7.9	2.2–29.2	*p* = 0.0018
November	9.0	2.2–36.2	*p* = 0.0020
December	25.4	3.1–211.1	*p* = 0.0028
**Season**			
Spring	4.0	1.9–8.1	*p* = 0.0002
Summer	1.9	1.0–3.4	*p* = 0.0407
Autumn	3.7	1.9–7.3	*p* = 0.0002
Winter	10.5	4.5–24.2	*p* < 0.0001
**Year**			
2016	3.8	2.3–6.3	*p* < 0.0001
2017	3.1	1.6–6.1	*p* = 0.0008
2018	5.0	2.6–9.6	*p* < 0.0001

* Reference: Wet market (odds ratio = 1).

**Table 4 foods-09-00264-t004:** Combined odds ratio analysis of *Salmonella* prevalence.

	Combined Odds Ratio ^a,^*	95 % Confidence Interval	*p* Value
Supermarket*Spring	1.0	0.47–2.29	*p* = 0.922
Supermarket*Summer	0.5	0.25–0.97	*p* = 0.040
Supermarket*Autumn	1.0	0.45–2.08	*p* = 0.940
Supermarket*Winter	2.8	4.5–24.2	*p* = 0.028
Year			
Supermarket*2016	1.0	0.55–1.83	*p* = 0.990
Supermarket*2017	0.8	0.38–1.73	*p* = 0.604
Supermarket*2018	1.3	0.63–2.74	*p* = 0.464

^a^ Combined odds ratio (Synergy factor) estimated as described by [38]. * Reference: Wet-market (odds ratio = 1).

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
