# Peer review of "Three-Year Longitudinal Study: Prevalence of Salmonella Enterica in Chicken Meat is Higher in Supermarkets than Wet Markets from Mexico"

_foods, 2020, doi:10.3390/foods9030264_

Round 1

Reviewer 1 Report

My biggest issue with the paper is biasing in the data collection. In the methods section the description of the carcass collection implies that the same number of carcasses (20) were collected each month for 3 years. However, the math is not correct and I figured that out after viewing table 1. It appears that a majority of the data was collected in 2016 (more than half). This fact is not discussed anywhere in the manuscript, misleading readers and may have led to a significant bias in data analysis. Bias in comparison of years, seasons, and with data collected from databases (human infection, weather conditions) most likely occurred. Given that the methods section description is not correct an accurate description is necessary in order to give a proper review.

My other question about the sampling is in regards to sampling consistency as this too can lead to a significant amount of bias - were the same wet market retailers and same retail markets sampled each month? It is very difficult to compare across markets due to multiple confounding factors including those that may occur on the farm and during transportation and holding. Although the authors indicated in the discussion section that refrigeration time for retail was expected to be longer and could have been a confounding factor, there are certainly many more that make a true comparison very difficult.

Other issues:

Please define wet-market

Grammar errors – Lines 36, 47, 66, 93, 100, 103, 121, 150, 173

I believe the authors had an issue with their bibliographic software as in many places there is an error message where reference numbers should be.

Line 37-39- Need more references for such a broad statement

Line 40 – I would not say 4 references constitutes extensive documentation throughout 4 continents.

Lines 130-131  -Why are the 4 seasons listed out? This is an odd way to write this sentence.

Materials and Methods : please give more information for the sample condition of carcasses from the live market. Were they plucked? Eviscerated? Evisceration can be a leading source of Salmonella on carcasses. Thus the absence of evisceration may have had an impact on the results but I did not see any information around this notion.

Lines 87 and 180- Most Journals strongly discourage or prohibit making “1st claims” which is done twice in this paper.

Conclusions: I feel that the conclusions are not supported by the data presented. The authors claim that this data provides a baseline for Salmonella prevalence and food safety risk in Mexico when this study was conducted in only 1 city. Furthermore, the assumption would be that the only source of Salmonella is raw chicken from retail and wet markets when in fact Salmonella as a foodborne pathogen can be sourced to many food items.  Lines 183 to 184 claims that Salmonella prevalence has increased in the past 3 years, but this cannot be supported for the same reasons (only one city, one food source, and potential sampling bias).

Reviewer 2 Report

I found this to be a well written manuscript of significance.  The authors have done a good job in Introducing the background and rationale for their study and clearly define their objectives.  Materials and methods are written concisely and clearly and in sufficient detail to allow reviewers to assess the soundness of their results and to allow us to repeat such a study if they so choose.  The results are presented succinctly and objectively and conclusions reached are supported by evidence.  A good job.  I have only one very minor comment.

Italicize Salmonella in table titles and Figure Legends.

Reviewer 3 Report

The manuscript by Regalado-Pineda and coworkers demonstrated the three-year longitudinal study on prevalence of Salmonella enterica in chicken meat obtained from supermarkets and wet-markets from Mexico. Overall, the article is well written. I suggest minor revision.

Introduction is very short and failed to establish connection of the present study with available literature. Authors should carefully work on it to establish connection of their proposed study with previous findings.   

In addition to the statistical analyses they have performed on their study, it would be more worthwhile to add the effect of combined independent variables (year, monthly interval, seasons, types of market) on dependent variables like prevalence and recovery etc.   
